# The Short Form of the Fonseca Anamnestic Index for the Screening of Temporomandibular Disorders: Validity and Reliability in a Spanish-Speaking Population

**DOI:** 10.3390/jcm10245858

**Published:** 2021-12-14

**Authors:** Noelia Zagalaz-Anula, Carmen María Sánchez-Torrelo, Faustino Acebal-Blanco, Roger Alonso-Royo, Alfonso Javier Ibáñez-Vera, Esteban Obrero-Gaitán, Daniel Rodríguez-Almagro, Rafael Lomas-Vega

**Affiliations:** 1Department of Health Sciences, Campus Las Lagunillas, University of Jaén, 23071 Jaén, Spain; nzagalaz@ujaen.es (N.Z.-A.); eobrero@ujaen.es (E.O.-G.); rlomas@ujaen.es (R.L.-V.); 2FisioMedic Clinic, 41701 Dos Hermanas, Spain; fisiomedic.dh@gmail.com (C.M.S.-T.); rar00032@red.ujaen.es (R.A.-R.); 3Service of Oral and Maxilofacial Surgery, Hospital Complex of Jaén, 23007 Jaén, Spain; faustino@faustinoacebal.com; 4Department of Nursing, Physiotherapy and Medicine, University of Almería, 04120 Almería, Spain; dra243@ual.es

**Keywords:** temporomandibular joint syndrome, reproducibility of results, surveys and questionnaires, validation study

## Abstract

The Short Form of the Fonseca Anamnestic Index (SFAI) is a simple and quick questionnaire used for screening temporomandibular disorders (TMDs). The present study aimed to validate the Spanish version of the SFAI in patients with TMDs. The study sample comprised 112 subjects (50 TMDs and 52 controls). Test–retest reliability, factorial validity, internal consistency, concurrent validity, and the SFAI’s ability to discriminate between TMDs subjects and healthy controls were analyzed using the Diagnostic Criteria for Temporomandibular Disorders (DC/TMD protocol) as the reference. Factor analysis showed a single factor that explained 63% of the total variance. Cronbach’s alpha was 0.849. The reliability of the items measured with the Kappa index showed values from 0.767 to 0.888. Test–retest reliability was substantial (intraclass correlation coefficient = 0.837). The total SFAI score showed a significant correlation with orofacial pain, vertigo, and neck disability measurements. For a cut-off point of >10 points, the SFAI showed a sensitivity of 78% and specificity of 78.85% at differentiating between TMDs patients and healthy subjects, with an area under the curve (AUC) of 0.852. The Spanish version of the SFAI is a valid and reliable instrument for diagnosing people with TMDs and shows generally good psychometric properties.

## 1. Introduction

Temporomandibular disorders (TMDs) are orofacial pain problems characterized by pain in the facial and mandibular structures [1]. The International Association for the Study of Pain (IASP) defines orofacial pain as “perceived pain in the face and/or oral cavity” caused by diseases or disorders of the nervous system, nearby structures, or distant structures [2]. The main characteristic of TMDs is pain in the temporomandibular joint (TMJ) area. However, it also affects cranial, cervical, and facial muscles, with limitation of mandibular movement and the presence of noises such as clicks and crackles during movement of the mandible [1].

TMD may affect up to 50% of the general population [3,4], producing a high burden on health care services. A correct diagnosis of TMDs must be established by anamnesis, physical examination, and, if necessary, diagnostic imaging tests [5]. This process is crucial to properly treat the affected person, as a wrong diagnosis leads to incorrect treatment and therefore has negative repercussions on the patient’s quality of life [6].

Some authors have defended the need to address TMDs in primary care, enabling better results for patients and maintaining the skill level and experience of specialist services in secondary care [7]. However, TMDs usually present concomitantly with other very prevalent disorders such as tinnitus [8], headache [9], or neck pain [10], and consequently can go unnoticed with resultant delay in health care. Therefore, TMD screening is essential for correctly establishing personalized attention.

The Short Form of the Fonseca Anamnestic Index (SFAI) [11] is a self-implemented test that was created and validated in 2018 in Brazil by Fernandes Pires and collaborators from the Fonseca Anamnestic Index (FAI) [12]. The SFAI has five items and is a quick and straightforward questionnaire used to evaluate TMD, providing the opportunity to optimize the diagnostic and screening process. Currently, only the original validation in Portuguese [11], built from the elimination of five questions from the original FAI, is available. The SFAI has shown a very good capacity to predict TMDs when compared to the main benchmark, the Diagnostics Criteria for Temporomandibular Disorder protocol (DC/TMD) [13]. The FAI was recently translated and cross-culturally adapted to Spanish [14], although to date, its short version has not been validated for use in the Spanish population despite its ease of use and application. As with the original version, the Spanish version of the SFAI was constructed by extracting five items from the ten-item version.

The objective of this study was to validate the SFAI in the Spanish population and assess its psychometric properties in patients with TMDs. This study hypothesized that the Spanish SFAI presents good psychometric properties, with a monofactorial structure, good internal consistency and acceptable reliability, moderate concurrent validity with respect to other diagnostic tools, and good diagnostic accuracy with respect to the gold standard, the DC/TMD protocol.

## 2. Materials and Methods

### 2.1. Participants

A cross-sectional questionnaire validation study was conducted to meet the objectives of this study. Ethics approval from the Research Ethics Committee of Jaen was obtained (Date: 26 September 2019; code: FonsecaUJA). This study was designed in accordance with the Declaration of Helsinki and the code of Good Researching Practices of University of Jaén, based on the applicable laws and regulations. All subjects participating in the study had to provide written informed consent. The sample selection was performed in the FisioMedic clinic (Dos Hermanas, Spain) among all those patients who attended the Physiotherapy, General Medicine, Traumatology, and Stomatology services between 25 May 2020 and 26 August 2020. Doctors of the different services informed potential participants about the study and derivated the subjects to the researchers for further information after. A researcher was in charge of recruitment by telephone interview after a first telephone call. To calculate the sample size, we followed the calculations obtained in studies with different types of samples that recommend a minimum of 20 subjects for reliability studies and a minimum of 40 subjects for concurrent validity studies [15]. The sample of subjects also had to meet the criterion of a minimum number of five and an optimal number of 10 subjects for each item of the instrument to guarantee factor validity and internal consistency analyzes [16]. As the tool has five items, a total of 50 patients were required.

Patients 18 years or older and diagnosed with pain-related TMD by the DC/TMD were considered eligible for this study. Severe neurological or psychiatric pathologies that could influence the correct completion of the questionnaires or data provided to researchers (such as dementia, Alzheimer’s, schizophrenia, Parkinson’s or amyotrophic lateral sclerosis) were considered as a reason for exclusion from the study. Patients under treatment with anti-depressants or opioids were also excluded. Furthermore, a sample of healthy subjects who did not meet the DC/TMD diagnostic criteria for TMDs was used to test the ability of the SFAI to discriminate between patients and healthy subjects.

### 2.2. Measurements

First, interviews were conducted with the participants to collect demographic data such as sex, age, educational level, work situation, height, weight, body mass index (BMI), smoking habits, alcohol intake, and physical activity level. The interviewer in charge of the clinical measurements was a physician with 20 years of experience and who was appropriately trained in the instruments used.

The DC/TMD examination protocol [17] is composed of the most widely used diagnostic criteria for temporomandibular disorders. This instrument presents three major components: A questionnaire on symptoms (assesses muscle and joint pain, type of bite, jaw movements, presence of headache in the last 30 days, opening pattern, joint noises, and blockages); a protocolized clinical examination; and the diagnostic algorithms. Finally, utilizing a diagnostic tree, the instrument differentiates between diagnostic categories (pain-related TMD, headache, intra-articular disorder, degenerative joint disorder, and/or subluxation). Neither the researcher in charge of the general assessment nor the participants were aware of the DC/TMD examination protocol results, which another researcher performed to avoid risk of bias.

The Fonseca Anamnestic Index (FAI) was developed and validated in 1992 [12]. This index is composed of 10 questions with three response options (0 = no; 5 = sometimes; and 10 = yes), with an overall score that ranges from 0 to 100. The FAI aims to evaluate the presence or absence of TMD symptoms and their severity, classifying them as mild, moderate, or severe. SFAI was obtained after factorial analysis, extracting items 1–3, 6, and 7 from the FAI [12]. The Spanish version of the SFAI was obtained by extracting the appropriate items from the Spanish version of the FAI (Appendix A), which was recently translated and cross-culturally adapted by Sánchez-Torrelo et al. [14].

To evaluate pain intensity, the Numeric Pain Rating Scale (NRS) was chosen. This is a self-implemented scale on pain intensity perception. The score ranges from 0 (total absence of pain) to 10 (maximum and extreme pain that the patient is capable of imagining), presented in an increasing manner from left to right; the subject would have to mark the answer considered with a cross [18]. In this study, the participants recorded separately on two independent NRS scales orofacial pain and neck pain.

The 12-Short Form Health Survey was chosen to assess health-related quality of life in this study. This is a simpler and quicker version of the SF-36, being self-administered and evaluating general quality of life from both a physical and an emotional point of view. It is composed of 12 questions that present a variable number of answers. The final score of the test is obtained by a statistical processing instrument that provides in a more exact way the value of the physical and mental summary scores with a value ranging from 0 to 100 [19].

The Dizziness Handicap Inventory (DHI) was used to measure dizziness and vertigo sensations. This questionnaire is a self-implemented scale used to identify lack of balance and vertigo conditions. It consists of 25 questions that can be answered as “no”, “sometimes”, or “yes”. This inventory identifies physical, functional, and emotional problems related to dizziness. Each dimension is composed of different questions distributed in random order throughout the test [20,21,22].

The Headache Impact Test (HIT-6) is a self-administered questionnaire on headaches. It contains six questions and five possible answers for each question: “always”, “very often”, “sometimes”, “rarely”, and “never”. A numerical score results from the sum of the answers. This questionnaire has been adapted and validated for peninsular Spanish speakers [23] and several other languages.

The Neck Disability Index (NDI) was used to measure disability produced by neck pain. This questionnaire consists of ten questions that can be answered on a six-point scale from least (0) to most disability (5). The final score is the sum of the answers, with a resulting range from 0 to 50. The final score is categorized as: “no disability” if the result is 0–4, “moderate disability” for the range 15–24, and “complete disability” for a score in the range 35–50 [24].

### 2.3. Statistical Analysis

SPSS 20.0 statistical package (SPSS Inc, Chicago, IL, USA) and MedCalc Statistical Software version 19.1.5 (MedCalc Software, Ostend, Belgium) were used for data management and analysis. Means and standard deviations for continuous variables and frequencies and percentages for categorical variables were calculated to describe the data. The Kolmogorov–Smirnov test was selected to verify the normality of the continuous variables, and the Levene test was used to test the homoscedasticity of the samples. To analyze possible differences between subjects and controls, the Student’s t test was used for continuous variables and the chi-square test for categorical variables. The confidence level was set at 95% (*p* < 0.05).

Factorial validity was evaluated by using principal component analysis (PCA) with Kaiser varimax rotation. Bartlett’s test and the Kaiser–Meyer–Olkin (KMO) measure of sampling adequacy was obtained [25].

Test–retest reliability was determined using the intraclass correlation coefficient (ICC 2,1 by Shrout and Fleiss) [26]. Reliability was considered poor (ICC < 0.40), moderate (ICC = 0.40–0.75), substantial (ICC = 0.75–0.90), or excellent (ICC > 0.90) [26]. To analyze the precision of the score, the standard error of measurement (SEM) was calculated as standard deviation (SD) at baseline (σbase) minus the square root of (1-Rxx), where Rxx is the reliability index (ICC) [27]. The reliability between the two measurements of each item was analyzed using the weighted Kappa coefficient [28]. The reliability was by agreement considered null (Kappa = 0.00), insignificant (Kappa 0.00 to 0.20), discreet (Kappa = 0.21–0.40), moderate (Kappa = 0.41 to 0.60), substantial (Kappa = 0.61 to 0.80), and almost perfect (Kappa = 0.81–1.00) [29]. Additionally, the MDC was calculated at a 95% confidence level (MDC95) as follows: MDC95 = 1.96 × σbase × $\sqrt{1}-\mathrm{ICG}$, where 1.96 is the z-value corresponding to the 95% confidence interval (MDC95). The MDC provides a good opportunity for translating the reliability index into units of change of the instrument. In addition, Bland–Altman plots were obtained to evaluate the limits of agreement [26].

Cronbach’s α coefficient was used to measure internal consistency. The α coefficient was considered poor if it was less than 0.70, and good if it was between 0.70 and 0.90; it was interpreted as indicating redundancy if α was greater than 0.90 [30].

Pearson’s correlation coefficient was selected for the concurrent validity analysis of the SFAI with the NDI, DHI, HIT-6, SF-12, and NRS. Correlation was considered poor for values below 0.30, moderate if it was between 0.30 and 0.50, and strong if it was higher than 0.50 [31].

The SFAI total score’s ability to discriminate between TMD and healthy subjects was analyzed using receiver operating characteristic (ROC) curves. Patients with or without TMDs were classified according to the DC/TMD protocol criteria, and the SFAI total score was used as the variable. The area under the curve (AUC) was obtained as a measure of the parameter’s ability to discriminate between subjects with TMDs and healthy controls. The AUC was considered statistically significant when the 0.5 value was not included between the 95% confidence interval [32]. The accuracy was considered low when AUC was between 0.5 and 0.7, good when between 0.7 and 0.9, and high when greater than 0.9 [33].

## 3. Results

The sample consisted of 102 subjects, 50 belonging to the group of patients with pain-related TMDs, 36 subjects with myofascial pain (Ia), 14 with myofascial pain with limited mouthopening (Ib), and 52 to the group of healthy controls (Figure 1). The sociodemographic characteristics of the total sample and the two groups of subjects are shown in Table 1.

The factor analysis showed a structure compatible with the existence of a single factor (Figure 2) that explained 63% of the total variance (Table 2). The KMO of 0.835 (X2 = 225.516, *p* < 0.001) turned out to be satisfactory; therefore, the sample was suitable to be analyzed by factor analysis.

The internal consistency analysis showed a good result with a Cronbach’s alpha of 0.849. In the analysis of the items, good correlations were found for each item with the total score, and the elimination of each one of the items slightly worsened the alpha value (Table 3).

The reliability of the items measured with the Weighted Kappa showed values from 0.767 to 0.888 (Table 4). The test–retest reliability of the global score yielded an ICC value of 0.837, which can be considered as substantial reliability. The limits of agreement obtained with the Bland and Altman analysis are shown in Figure 3. A SEM of 3.47 and a MDC of 6.79 were found.

The receiver operating characteristic (ROC) curve analysis yielded an AUC value of 0.852 (95% CI 0.768 to 0.915, *p* < 0.001), which can be considered as showing a good ability to discriminate between patients with TMDs and controls (Figure 4). For a cut-off point of >10 points on the scale, the sensitivity was 78% and the specificity was 78.85% (Table 5). Given the differences in the proportion of men and women in both groups of subjects and controls, an analysis was made by sex, finding an AUC of 0.847 (standard error = 0.046) in women, and an AUC of 0.917 (standard error = 0.057) in men. The comparison between both ROC curves was not statistically significant (*p* = 0.339).

The Spanish version of the SFAI showed good and significant correlation with the other indices of TMD assessment and with measures of orofacial pain, neck disability, and the evaluation of vertigo in the concurrent validity analysis. Nevertheless, the correlations with NRS cervical pain, SF-12 PCS, and HIT-6 can be considered as “poor” and in the case of the PCS statistical significance was not even reached (Table 6).

## 4. Discussion

The present study assessed the characteristics of the SFAI, the short version of the FAI, to evaluate TMDs in a Spanish-speaking population. The results report that this test is valid and reliable, assesses severity, and shows the ability to differentiate between subjects affected or not by TMDs. The mean time to complete the questionnaire was between one and two minutes, making SFAI an ideal tool for use in primary care centers.

Our sample of subjects with TMDs had a higher proportion of women than men, in line with evidence reported in the scientific literature. We believe that this sample is representative of the TMD population, and although the control group differed in the proportion of men and women with respect to the group of subjects, the analyses of ROC curves by subgroups have provided similar information.

Fernandes Pires et al. developed and validated the SFAI for the first time in the Brazilian population in 2017 [11] with a sample of 123 women, of which 57 suffered from myogenic TMDs and 66 did not suffer from TMD symptoms. The patients were evaluated using the Research Diagnostic Criteria for TMDs, and responded to the SFAI test twice within a week. The results showed excellent reliability with an ICC of 0.95 in all items and 0.98 in the total score of the index. The sensitivity of the test was 0.97, with a cut-off point of 17.5 points.

In general, the results of the previous study appear better than those obtained in ours. This may be because Fernandes Pires’ sample set contained patients who were on average 25 years younger than ours (47–48 years) and was composed only of women with muscle disorders [11], while ours also included men. The original SFAI validation study results can be extrapolated to other young populations with a greater capacity to respond to simple instructions. In contrast, our study presents more modest but perhaps more realistic results applicable to a broader population in terms of age and sex. Regarding the test–retest reliability, the data from the original validation study are also exceptionally high compared to ours. This may be because the authors used the ICC to measure the reproducibility of the items, which is highly questionable as these items have an ordinal response and the ICC should be applied to quantitative data. In our work, we used the Weighted Kappa Index by quadratic weights, which is the appropriate measure for the response scale of the items.

Sánchez-Torrelo et al. validated the Spanish version of the FAI in 2020 [14]. Compared with the standard version of the FAI in Spanish, the short version is completed in half the time. Nonetheless, the two scores are essentially equivalent and presenting a very high correlation between the two (r = 0.88). The factorial validity of the standard version in Spanish showed a structure with three factors, the first of which practically reproduces the content of the SFAI, except for the inclusion of an additional item. We opted, however, to keep the item structure of the original instrument in Portuguese. As in our study, the test–retest reliability of the items ranged from substantial to almost perfect, although the standard version of the FAI showed a reliability of the total score that could be considered excellent, compared to a lower reliability obtained in our study for the short version that could be considered substantial. The SEM was 6.28, and the MDC was 12.31 for the standard version, which can be considered equivalent to the data obtained in our study because the short scale has half as many items as the standard version. The internal consistency showed a Cronbach’s α of 0.784, which is somewhat lower than the alpha obtained in our study for the short version. Concurrent validity showed a low correlation with the HIT-6 and PCS-SF12 tests, moderate correlation with the DHI, SF12, NDI, and cervical pain NRS, and high correlation with the NRS for orofacial pain. The ROC curve yielded an AUC of 0.865, a sensitivity of 82%, a specificity of 78%, and a cut-off point >35 points. Regarding the ability to discriminate between subjects with and without TMDs, the predictive values were very similar, with both sensitivity and specificity close to 80%. In the case of the standard version, the cut-off point for the diagnosis of TMDs was more than 35 points. In the short version in Spanish, this cut-off point was 10 points, which is equivalent to being able to screen any patient who answers positively to any of the items as TMD and being able to rule out the disease only when the global score is zero.

Our study has several limitations. First, the sample size is tight, although it is similar to the original valuation study. There were slight differences between the control and comparison groups, justifiable by the higher prevalence of TMD among women. Moreover, despite the wide age range used to increase generalizability, it could also be interpreted as a limitation. However, few comparable studies exist, so we cannot reach further conclusions in this regard. It should also be noted that the evaluators, although highly experienced in the subject, were not calibrated. Furthermore, the sample set of subjects was obtained in a very specific geographic area, so it would be desirable to analyze the results in broader socio-cultural areas. Finally, and although our study analyzed the most common psychometric properties, other properties of interest remain unknown such as whether sensitivity to change or the ability to discriminate vary between different types of TMDs.

## 5. Conclusions

The Spanish version of the SFAI is a valid and reliable instrument for diagnosing people with pain-related TMDs, with a test–retest reliability between “moderate” and “substantial”, good internal consistency, and a good ability to differentiate between affected and unaffected subjects. In addition, the correlation between SFAI and other specific evaluation instruments for TMD is strong.

## Figures and Tables

**Figure 1 jcm-10-05858-f001:**
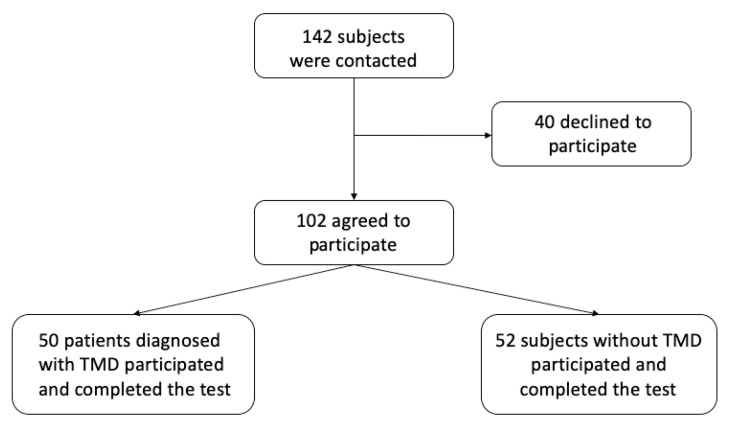
Flow diagram of the participants.

**Figure 2 jcm-10-05858-f002:**
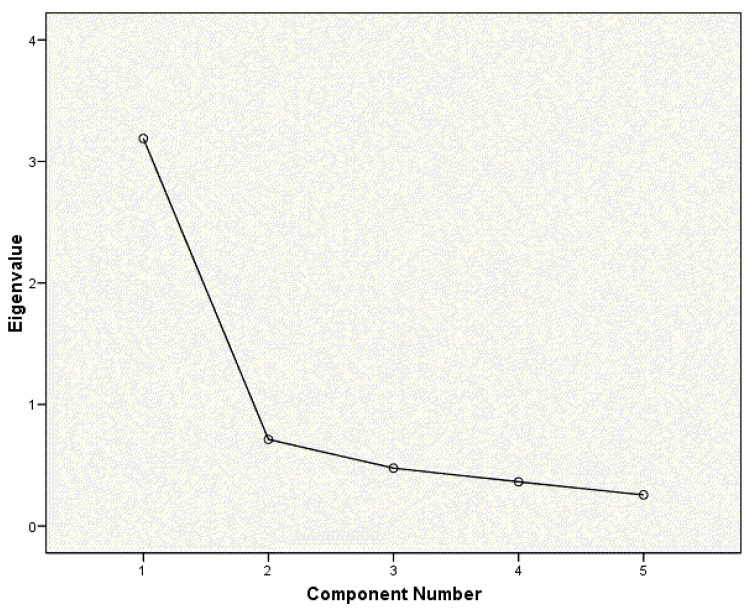
Scree plot for factorial analysis.

**Figure 3 jcm-10-05858-f003:**
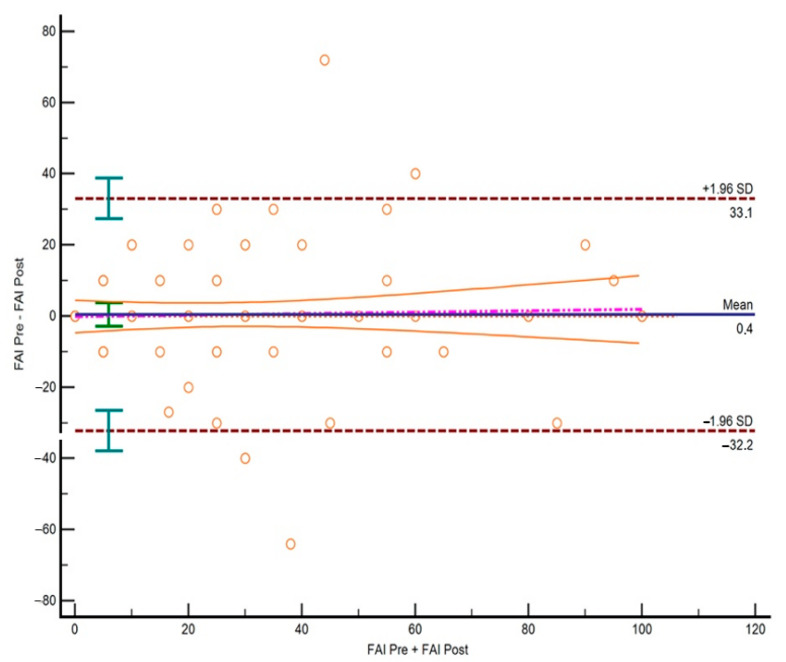
Bland–Altman plot.

**Figure 4 jcm-10-05858-f004:**
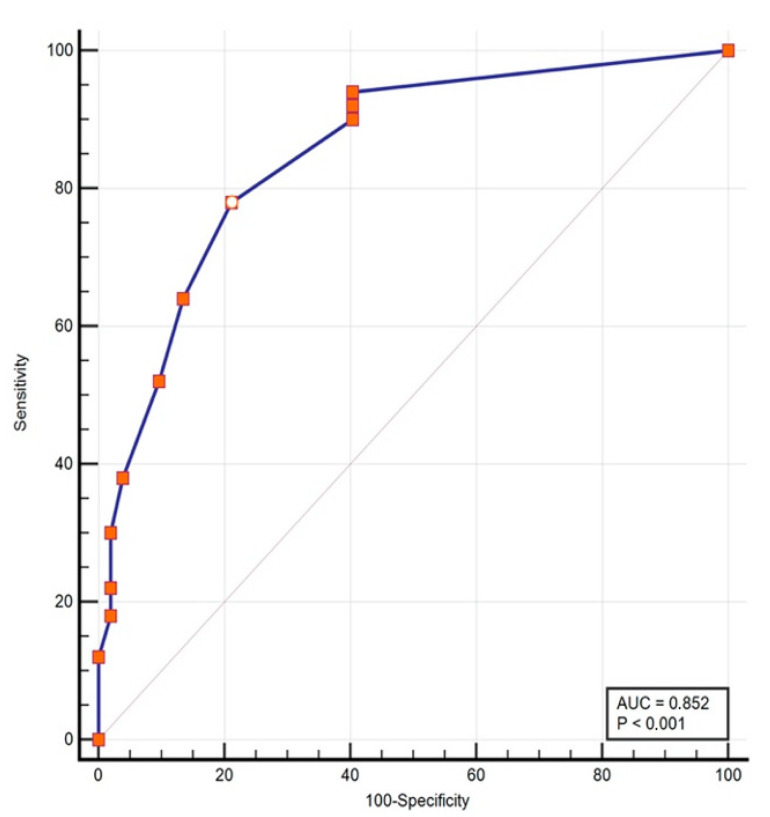
Predictive values in the ROC curve analysis.

**Table 1 jcm-10-05858-t001:** Sociodemographic variables.

Variables		All Participants		No TMDs		TMDs		Differences
Continuous		*n* = 102		*n* = 52		*n* = 50		*p*-Value
		**Mean**	**SD**	**Mean**	**SD**	**Mean**	**SD**	
Age (Years)		47.07	13.79	48.85	14.67	45.18	12.67	0.184
Weight (Kg)		73.43	16.56	76.43	17.11	70.31	15.53	0.050
Height (m)		1.62	0.09	1.65	0.10	1.60	0.08	0.005
Body Mass Index		27.94	6.84	28.09	6.34	27.79	7.39	0.665
**Categorical**		**F**	**%**	**F**	**%**	**F**	**%**	
Gender	Male	25	24.5%	21	40.4%	4	8.0%	<0.001
	Female	77	75.5%	31	59.6%	46	92.0%	
Academic level	Primary	20	19.6%	13	25.0%	7	14.0%	0.036
	Secondary	51	50.0%	29	55.8%	22	44.0%	
	University	31	30.4%	10	19.2%	21	42.0%	
Physical Activity	Yes	62	60.8%	34	65.4%	28	56.0%	0.438
	No	40	39.2%	18	34.6%	22	44.0%	
Smoking habits	No smoker	66	64.7%	31	59.6%	35	70.0%	0.731
	Smoker	14	13.7%	8	15.4%	6	12.0%	
	Occasional smoker	9	8.8%	5	9.6%	4	8.0%	
	Former smoker	13	12.7%	8	15.4%	5	10.0%	
Alcohol habits	No drinker	37	36.3%	19	36.5%	18	36.0%	0.937
	Habitual drinker	7	6.9%	4	7.7%	3	6.0%	
	Occasional	58	56.9%	29	55.8%	29	58.0%	
Economic Level	<20.000 €	63	61.8%	33	63.5%	30	60.0%	0.719
	>20.000 €	39	38.2%	19	36.5%	20	40.0%	

TMDs: Temporomandibular disorders; SD: Standard deviation; F: Frequency.

**Table 2 jcm-10-05858-t002:** Total variance explained by the factor analysis.

Component	Initial Eigenvalues			Extraction Sums of Squared Loadings		
	Total	% of Variance ^a^	Cumulative % ^b^	Total	% of Variance ^a^	Cumulative % ^b^
Item 1	3.189	63.784	63.784	3.189	63.784	63.784
Item 2	0.713	14.266	78.050			
Item 3	0.477	9.537	87.587			
Item 4	0.364	7.285	94.871			
Item 5	0.256	5.129	100.000			

^a^ Percentage of variance that explains each factor of the questionnaire structure. ^b^ Total percentage of variance explained jointly by the factors that compose the questionnaire structure.

**Table 3 jcm-10-05858-t003:** Item analysis of the Spanish Version of the Short Form of the Fonseca Anamnestic Index (SFAI).

	Mean of the Scale If the Element Is Deleted	Scale Variance If the Element Is Removed	Corrected Total-Element Correlation	Multiple Squared Correlation	Cronbach’s Alpha If Element Is Deleted ^a^
Item 1	9.74	6.137	0.681	0.549	0.816
Item 2	9.77	5.701	0.761	0.644	0.794
Item 3	9.97	5.415	0.707	0.556	0.804
Item 4	10.15	5.513	0.620	0.393	0.830
Item 5	10.06	5.739	0.568	0.345	0.844

^a^ Cronbach’s alpha value if the item is deleted from the analysis. Items 1–5: Questions of the Short Form of the Fonseca Anamnestic Index.

**Table 4 jcm-10-05858-t004:** Test–retest reliability of the items and the Short Form of the Fonseca Anamnestic Index total score.

ITEM	Weighted Kappa	Lower Bound	Upper Bound	Reliability
Item 1	0.732	0.558	0.906	Substantial
Item 2	0.746	0.583	0.909	Substantial
Item 3	0.898	0.838	0.957	Almost perfect
Item 4	0.790	0.700	0.881	Substantial
Item 5	0.682	0.547	0.817	Substantial
Total score ^a^	0.837	0.767	0.888	Substantial

^a^ Intraclass correlation coefficient (ICC) value for the overall Short Form of the Fonseca Anamnestic Index score. Items 1–5: Questions of the Short Form of the Fonseca Anamnestic Index.

**Table 5 jcm-10-05858-t005:** Predictive values of the Short Form of the Fonseca Anamnestic Index to diagnose cases of Temporomandibular disorder.

Criterion	Sensitivity		95% CI	Specificity		95% CI	
>10	78.00%		64.0–88.5	78.85%		65.3–88.9	
**+LR**	**95% CI**	**−LR**	**95% CI**	**+PV**	**95% CI**	**−PV**	**95% CI**
3.69	2.1–6.4	0.28	0.2–0.5	78.0	67.3–85.9	78.8	68.5–86.5

95% CI: 95% confidence interval; +LR: positive likelihood ratio; −LR: negative likelihood ratio; +PV: positive predictive value; −PV: negative predictive value.

**Table 6 jcm-10-05858-t006:** R coefficient between SFAI and other variables.

Variable	r Coefficient	*p*-Value	Correlation
FAI	0.876	<0.001	Strong
NRS orofacial pain	0.660	<0.001	Strong
NRS cervical pain	0.287	0.003	Poor
DHI functional	0.348	<0.001	Moderate
DHI emotional	0.404	<0.001	Moderate
DHI physical	0.409	<0.001	Moderate
SF-12 PCS	0.023	0.821	Poor
SF-12 MCS	0.353	<0.001	Moderate
NDI	0.360	<0.001	Moderate
HIT-6	0.229	0.021	Poor

FAI: Fonseca Anamnestic Index; NRS: Numeric Pain Rating Scale; NDI: Neck Disability Index; SF-12 PCS: Short-Form Health Survey Physical Component Summary; SF-12 MCS: Short-Form Health Survey Mental Component Summary; DHI: Dizziness Handicap Inventory; HIT-6: The Headache Impact Test.

## Data Availability

The data in this study are available under request to the corresponding author.

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
