# Peer review of "The Short Form of the Fonseca Anamnestic Index for the Screening of Temporomandibular Disorders: Validity and Reliability in a Spanish-Speaking Population"

_jcm, 2021, doi:10.3390/jcm10245858_

Round 1

Reviewer 1 Report

REVIEW

Overall comments

The authors try to make a valuable contribution by trying to validate a TMD diagnostic questionnaire (SFAI) initially developed for the Portuguese language into Spanish. However, a thorough review must be made  due to possible patient/control selection biases (false negatives) and lack of control for confounding variables (please see review below). Therefore, the study must be reconsidered after major revisions.

Materials and Methods

Participants

Line 84: The sample size calculation was based in previous studies and recommendations, and not in a sample size calculation; therefore, it is a convenience sample. Please describe the sample size calculation with the appropriate parameters.

Line 87: The selection of the TMD cases was performed using the DC/TMD, were the controls also selected using the same DC/TMD clinical history and examination performed in the TMD cases?

Line 89: Patients 18 years old or older were selected, but this will include elderly population usually not affected by TMD. I believe that the age range between 18 and 50 years old (young adults) would have been selected, giving external validity to the results.

Line 90: Which severe neurological and psychiatric pathologies were used as excluding criteria, please describe? How about medication affecting the central nervous system, and pain perception, such as anti-depressants?

Line 92: same comments as in line 87 (see above)

Line 108: Was the examiner in charge of the clinical examination blinded to the results of the questionnaire on symptoms? Were these examiners calibrated in anyway, trained is not enough?

Results

Line 197: Table 1: The mean age of both TMD cases and controls seems to be higher than reported in specialized literature (between 20 to 40 years old), please see comments on line 89. In addition, the possible socio-demographic differences seen in Table 1 between cases and controls (i.e., age, weight, particularly the gender proportions - 92% versus 59.6%, and others) must be tested statistically to see if there are differences. When statistically differences are found, they must be taken into account in the multivariate analysis. I believe that possible confounders were neither identified nor analyzed.

No results were presented regarding the diagnostic TMD groups in which the selected TMD cases were diagnosed: muscle disorders, disc displacements, arthralgia, etc.

Discussion

Line 275: The authors identified a higher proportion of women in the TMD cases, but make no reference about the huge differences between cases and the control group, or the measures to control this variable in the analysis stage.

Line 279: According to the authors: “No notable differences were found for to age, weight, height, BMI, average family income, or alcohol consumption”. Again, how can the authors possibly make that assumption, considering that no statistical test was used to compare TMD cases versus controls in Table I. Again, if detected, this identified confounder should have been handled in the statistical analysis.

Line 294: The authors stated: “In general, the results of the previous study appear better than those obtained in ours. This may be because Fernandes Pires' sample set contained patients who were on average 25 years younger than ours (47–48 years) and was composed only of women with muscle disorders (11), while ours also included men.” As already pointed out in previous comments, considering that the authors were trying to replicated the SFAI results in the Portuguese language (Fernandes previous study), it would have been more coherent to select the same age range, gender distribution, and TMD diagnostic groups of the previous reference study. Why did the authors make a different selection of cases and controls?

THIS IS THE REVIEW!

Author Response

First of all, we want to thank the reviewer for the comments and suggestions, which have allowed us to improve the manuscript.

Overall comments

The authors try to make a valuable contribution by trying to validate a TMD diagnostic questionnaire (SFAI) initially developed for the Portuguese language into Spanish. However, a thorough review must be made due to possible patient/control selection biases (false negatives) and lack of control for confounding variables (please see review below). Therefore, the study must be reconsidered after major revisions.

Materials and Methods

Participants

Line 84: The sample size calculation was based in previous studies and recommendations, and not in a sample size calculation; therefore, it is a convenience sample. Please describe the sample size calculation with the appropriate parameters.

The authors thank the reviewer for this concern. There are numerous techniques that can be used for the calculation of the sample size oriented according to the statistical analysis that affects the main objective of an investigation. However, in validation studies, the sample must guarantee that factor validity, internal consistency, validity and reliability analyses can be carried out with guarantees. We have cited works that guide towards the minimum sample necessary for factor analysis and internal consistency, we have also revised the wording offering additional criteria that were used to obtain the sample and that are related to reliability and validity (lines 89-93). The authors believe that some criteria, accepted by the scientific literature, and which cover the sample needs of most of the psychometric properties analysed, should be adequate.

Line 87: The selection of the TMD cases was performed using the DC/TMD, were the controls also selected using the same DC/TMD clinical history and examination performed in the TMD cases?

Thank you for this comment. Yes, healthy controls were selected using the same DC/TMD clinical history and examination. This has been specified in lines 100-101.

Line 89: Patients 18 years old or older were selected, but this will include elderly population usually not affected by TMD. I believe that the age range between 18 and 50 years old (young adults) would have been selected, giving external validity to the results.

Thank you very much for this point. Our aim was to validate an easy, quick to use and generalizable tool for the diagnosis of temporomandibular disorders. According to the literature available, the percentage of elderly population with temporomandibular disorders is among 61% (https://pubmed.ncbi.nlm.nih.gov/24714862/) and 50.5% (https://pubmed.ncbi.nlm.nih.gov/28177060/). Although it is true that prevalence seems to get reduced with the age (https://pubmed.ncbi.nlm.nih.gov/29719041/), we consider those data of prevalence as important, so the inclusion of elderly participants was needed to increase generalizability.

Line 90: Which severe neurological and psychiatric pathologies were used as excluding criteria, please describe? How about medication affecting the central nervous system, and pain perception, such as anti-depressants?

We want to thank the reviewer for this suggestion. This information has been added in lines 95-97.

Line 92: same comments as in line 87 (see above)

Thank you for the comment, corrected in line 96.

Line 108: Was the examiner in charge of the clinical examination blinded to the results of the questionnaire on symptoms? Were these examiners calibrated in anyway, trained is not enough?

Thank you for this point. There was a unique examiner in charge of applying the DC/TMD, who was different from the one in charge of the other questionnaires, who at the same time, was blinded to the DC/TMD results. The sentences have been rewritten to clarify that there was only one researcher in the general assessment, so no calibration was needed among researchers. This information is available at lines 116-117.

Results

Line 197: Table 1: The mean age of both TMD cases and controls seems to be higher than reported in specialized literature (between 20 to 40 years old), please see comments on line 89. In addition, the possible socio-demographic differences seen in Table 1 between cases and controls (i.e., age, weight, particularly the gender proportions - 92% versus 59.6%, and others) must be tested statistically to see if there are differences. When statistically differences are found, they must be taken into account in the multivariate analysis. I believe that possible confounders were neither identified nor analyzed.

Thank you very much for this indication. To respond to the reviewer's concern, we first performed the group difference test using the t student test for continuous variables and the Chi-square test for categorical variables, and we have recorded these differences in Table 1. Also, the first paragraph of statistical analysis has been modified to record the techniques used in this analysis (lines 164-165). The fundamental difference between both groups was the unequal proportion of men and women, and secondary differences to this, such as some morphological difference or level of studies. As these differences could affect the analysis comparing both groups (ROC curve analysis), we have performed the ROC curve analysis for both sexes separately and compared the result with a specific test available in the MedCalc software. The result of this analysis by sex has been recorded in the corresponding subsection (lines 255-258).

No results were presented regarding the diagnostic TMD groups in which the selected TMD cases were diagnosed: muscle disorders, disc displacements, arthralgia, etc.

Thank you for this suggestion. Our sample consisted of patients with myogenous TMD. We have stated this in the first paragraph of results (lines 202-203).

Discussion

Line 275: The authors identified a higher proportion of women in the TMD cases, but make no reference about the huge differences between cases and the control group, or the measures to control this variable in the analysis stage.

Thank you for the question. As mentioned in a previous section, the gender differences between both groups affect only the ROC curve analysis. We have repeated this analysis in each subpopulation of men and women and the results have been specified in the corresponding section (lines 255-258).

Line 279: According to the authors: “No notable differences were found for to age, weight, height, BMI, average family income, or alcohol consumption”. Again, how can the authors possibly make that assumption, considering that no statistical test was used to compare TMD cases versus controls in Table I. Again, if detected, this identified confounder should have been handled in the statistical analysis.

As previously indicated, the differences between both groups are relevant in the ROC Curve analysis. The differences between subjects and controls are significant in terms of the proportion of men and women, which implies that there are some secondary differences to the previous one in some sociodemographic variable such as weight. We have carried out the analysis of between-group differences of subjects and controls and we have stated this in Table 1. We have also carried out the analyses by groups of subjects based on sex and they have been recorded in the results section.

Line 294: The authors stated: “In general, the results of the previous study appear better than those obtained in ours. This may be because Fernandes Pires' sample set contained patients who were on average 25 years younger than ours (47–48 years) and was composed only of women with muscle disorders (11), while ours also included men.” As already pointed out in previous comments, considering that the authors were trying to replicated the SFAI results in the Portuguese language (Fernandes previous study), it would have been more coherent to select the same age range, gender distribution, and TMD diagnostic groups of the previous reference study. Why did the authors make a different selection of cases and controls?

Thank you very much for the comment. Although we understand your reasoning, our aim was to validate the SFAI for Spanish population rather than replicate the one of Portuguese language. Despite we consider this last a good work, our aim was to obtain a more generalizable validation, useful for a wider range of ages and groups. That is the reason why we used a wider sample.

THIS IS THE REVIEW!

Thank you very much for it.

Reviewer 2 Report

Dear authors, 

Congratulations on your work. In the attached document you can find some considerations to improve your paper.

Best regards.

Author Response

The present study analyses the psychometric properties of a short-form of the Fonseca Anamnestic Index, which is a recently used tool for the screening of temporomandibular disorders. The methodology used is adequate, the measurement tools used are acceptably explained, the analysis adequate. The study is one of the first validations of the instrument for its use in a language other than that of its creation. The results of this study will be useful to promote its use among physicians, dentists or physical therapists. However, the study presents some opportunities for improvement, with inconsistencies and writing errors.

Introduction
- Pag Lines 59-61: please re-write the sentence as “and” is not properly used.

Thank you very much for your comment, it has been re-written (line 60).

- Pag. Lines 65-66: please check the structure of this sentence, as they may be linked.

Thank you for the suggestion, it has been re-written (line 66).

Methods
- Although the analysis of psychometric properties is very extensive, the manuscript would gain strength if the authors had responsiveness results.

Thank you very much for this point. We agree that responsiveness has not being considered in this study, reason why this point was noticed in limitation sections as “sensitivity to change” (line 337).

- Although the term "clinimetric properties" is sometimes used, the term "psychometric properties" is more common. Please consider changing it throughout the text.

Thank you for the suggestion. “clinimetric” has been replaced by “psychometric” in the whole text.

- Pag. Lines 83-86: How were the participants informed about recruitment? A doctor of each of the services previously informed them about the possibility of participating or just facilitated the contact to the researchers?

Thank you for this point. This information has been added in lines 86-88.

- Page 2, line 91. If 50 patients are necessary for factor analysis and internal consistency, why were 52 healthy subjects selected? Consider merging the sentence on line 90-91 with the idea expressed on lines 95-96.

Thanks to the reviewer for his suggestion. We have eliminated the sentence on lines 87-88 that gave redundant information with those expressed in the next paragraph.

- Page 2, line 94. The eligibility criteria are not clear. "Severe neurological or psychiatric pathologies ..." Could the authors be more specific?

The authors want to thank the reviewer for this suggestion. This information has been added in lines 97-98.

- Pag. 3 Lines 119 and 120: both sentences begin the same: “The Spanish version of SFAI”, please check.

Thank you very much for this point. It has been solved in lines 124-125.

- Page 4, line 166. Could you place a reference here that justifies this categorization of the test-retest reliability?

Thanks for the suggestion. An appropriate reference has been added (line 169).

- Pag 4, line 183. In this section the authors classify the correlation as moderate or strong, but in the corresponding table, there are also values classified as "poor" that are not defined here. Please do not take this matter for granted.

Thank you for this observation. We have included the explanation in lines 185-186.

Results
- Page 4, line 197. Adverse effects are assumed to refer to a treatment. I don't understand the meaning of "Adverse Effects" in an observational study. I don't find much sense in this phrase.

Thank you for this observation. The referred sentence has been deleted.

- Page 7 line 231. The reliability index that I know is usually called in English "Weighted Kappa" and not "Kappa Index by weights". Please correct it.

The authors totally agree the reviewer. "Kappa Index by weights" has been replaced by "Weighted Kappa" (line 233).

- In Table 4, the Kappa value of the items does not correspond to the category expressed in the statistical analysis section. Please correct it.

Thanks for the observation. It has been corrected in Table 4 (line 239).

- Page 9, line 266. The content of this sentence does not agree with the content of table 6 to which it refers. Please clarify which information is correct.

Many thanks to the reviewer for this suggestion. We have corrected the error in the last sentence of the results (lines 278-279).

Discussion

-Line 275: please remove “Authors” as first word of Discussion.

Thank you for the observation, it has been removed.

- Page 10, line 283. There is a typographical error. However, this second paragraph discusses information irrelevant to the purpose of the study. From my point of view it should be eliminated, unless it is wanted to warn that the possible differences in the morphological variables may be due to a greater presence of women in the group of patients.

Thank you for the suggestion. The typographical error has been solved. Regarding the second paragraph, we have eliminated the irrelevant information, giving a new wording to the paragraph.

- Paragraph three of the discussion is a brief repetition of the results without providing additional information. From my point of view, this paragraph does not add value to the discussion and should be deleted.

Thank you very much for the suggestion. The referred paragraph has been deleted.

- Page 10, line 306. The word "data" appears twice. Please, delete the first.

Thank you for this point, it has been re-written.

- Page 10, line 310. The correct term is "Weighted Kappa index". Please modify it.

The authors agree with the change suggested, so it has been solved.

- Page 10, line 312. The article by Sánchez-Torrelo that appears in the bibliography is not from 2019 but from 2020. Please, modify it. Additionally, the information offered between lines 313-318 is a meaningless enumeration of the results of Sánchez-Torrelo et al. Please remove these phrases. If you have to discuss the results of the short and standard versions of the FAI, you can do it part by part giving meaning to the comparison of each property, but it does not make sense to list results from other works.

Thank you for this comment. The year has been changed to 2020 (line 314). Additionally, we have redrafted the paragraph trying to really discuss the similarities and differences in the psychometric properties of both instruments.

-Authors should add to supplementary materials the questionnaire to facilitate its clinical and research use.

Thank you very much for the suggestion. The questionnaire has been included as supplementary material (lines 389-390)

Reviewer 3 Report

I appreciate the work put into the research and writing the manuscript. It is a pity that the described questionnaire cannot be tested in all countries, but only in those where Spanish is spoken. Due to the large number of acronyms cited are quoted, I propose to introduce a summary of Abbreviations to the manuscript.

Author Response

I appreciate the work put into the research and writing the manuscript. It is a pity that the described questionnaire cannot be tested in all countries, but only in those where Spanish is spoken. Due to the large number of acronyms cited are quoted, I propose to introduce a summary of Abbreviations to the manuscript.

Thank you for the suggestion. A summary of Abbreviations has been added at the end of the manuscript (lines 369-388).

Round 2

Reviewer 1 Report

REVIEW

Overall comments

(PREVIOUS REVIEW) The authors try to make a valuable contribution by trying to validate a TMD diagnostic questionnaire (SFAI) initially developed for the Portuguese language into Spanish. However, a thorough review must be made prior to publication due to possible patient/control selection biases (false negatives) and lack of control for confounding variables (please see review below). Therefore, the study must be reconsidered after major revisions.

(PRESENT REVIEW) This version is much improved as compared to the previous one and most points have been clarified. This article is now acceptable for publication with few minor corrections, no need for new review.

Materials and Methods

Participants

(PREVIOUS REVIEW) Line 84: The sample size calculation was based in previous studies and recommendations, and not in a sample size calculation; therefore, it is a convenience sample. Please describe the sample size calculation with the appropriate parameters.

(PRESENT REVIEW) Line 89: Despite not providing a sample size calculation, the authors fundamented their sample sizes using a recent and valid literature, which is acceptable.

(PREVIOUS REVIEW) Line 87: The selection of the TMD cases was performed using the DC/TMD, were the controls also selected using the same DC/TMD clinical history and examination performed in the TMD cases?

(PRESENT REVIEW) Line 100: In this version, it is clear that the controls also underwent the DC/TMD, which was not clear in the previous version.

(PREVIOUS REVIEW) Line 89: Patients 18 years old or older were selected, but this will include elderly population usually not affected by TMD. I believe that the age range between 18 and 50 years old (young adults) would have been selected, giving external validity to the results.

(PRESENT REVIEW) Line 95: I believe that a statement about the large age range should be included in the discussion section as a limitation of the study, considering that the sample has already been selected and cannot be changed (minor correction).

(PREVIOUS REVIEW) Line 90: Which severe neurological and psychiatric pathologies were used as excluding criteria, please describe? How about medication affecting the central nervous system, and pain perception, such as anti-depressants?

(PRESENT REVIEW) Line 98: In this version, the neurological and psychiatric pathologies and the medication used which affected the CNS were described.

(PREVIOUS REVIEW) Line 92: same comments as in line 87 (see above)

(PREVIOUS REVIEW) Line 108: Was the examiner in charge of the clinical examination blinded to the results of the questionnaire on symptoms? Were these examiners calibrated in anyway, trained is not enough?

(PRESENT REVIEW) Line 116: In this version, it is clear that the clinical examiner was blinded to the questionnaire on symptoms of TMD. However, the examiners were not calibrated, and another statement in the discussion should be added (minor correction) as a limitation of the study.

Results

(PREVIOUS REVIEW) Line 197: Table 1: The mean age of both TMD cases and controls seems to be higher than reported in specialized literature (between 20 to 40 years old), please see comments on line 89. In addition, the possible socio-demographic differences seen in Table 1 between cases and controls (i.e., age, weight, particularly the gender proportions - 92% versus 59.6%, and others) must be tested statistically to see if there are differences. When statistically differences are found, they must be taken into account in the multivariate analysis. I believe that possible confounders were neither identified nor analyzed.

(PRESENT REVIEW) Considering that this is a reliability and validity study, I believe the control of confounders is not essencial, so it is acceptable at the present form.

(PREVIOUS REVIEW) No results were presented regarding the diagnostic TMD groups in which the selected TMD cases were diagnosed: muscle disorders, disc displacements, arthralgia, etc.

(PRESENT REVIEW) Line 201: It is clear now in this version that the diagnostic TMD groups were Ia and Ib of the DC/TMD.

Discussion

(PREVIOUS REVIEW) Line 275: The authors identified a higher proportion of women in the TMD cases, but make no reference about the huge differences between cases and the control group, or the measures to control this variable in the analysis stage.

(PRESENT REVIEW) Line 294: The authors in this version the authors left clear this limitation of the study regarding the gender differences between cases and controls, but again, considering that this a reliability and validity study, it is not also essencial.

(PREVIOUS REVIEW) Line 279: According to the authors: “No notable differences were found for to age, weight, height, BMI, average family income, or alcohol consumption”. Again, how can the authors possibly make that assumption, considering that no statistical test was used to compare TMD cases versus controls in Table I. Again, if detected, this identified confounder should have been handled in the statistical analysis.

(PRESENT REVIEW) Table 1: The authors added in this version the statistical tests and the p-value. Once more, considering that this is not a etiology study, this limitation does not affect the results.

(PREVIOUS REVIEW) Line 294: The authors stated: “In general, the results of the previous study appear better than those obtained in ours. This may be because Fernandes Pires' sample set contained patients who were on average 25 years younger than ours (47–48 years) and was composed only of women with muscle disorders (11), while ours also included men.” As already pointed out in previous comments, considering that the authors were trying to replicated the SFAI results in the Portuguese language (Fernandes previous study), it would have been more coherent to select the same age range, gender distribution, and TMD diagnostic groups of the previous reference study. Why did the authors make a different selection of cases and controls?

(PRESENT REVIEW) Line 315: In this version, the authors pointed out the differences between the two studies (Fernandes and this one), which was not clear in the previous version.

THIS IS THE REVIEW!

Author Response

Overall comments

(PREVIOUS REVIEW) The authors try to make a valuable contribution by trying to validate a TMD diagnostic questionnaire (SFAI) initially developed for the Portuguese language into Spanish. However, a thorough review must be made prior to publication due to possible patient/control selection biases (false negatives) and lack of control for confounding variables (please see review below). Therefore, the study must be reconsidered after major revisions.

(PRESENT REVIEW) This version is much improved as compared to the previous one and most points have been clarified. This article is now acceptable for publication with few minor corrections, no need for new review.

Thank you very much.

Materials and Methods

Participants

(PREVIOUS REVIEW) Line 84: The sample size calculation was based in previous studies and recommendations, and not in a sample size calculation; therefore, it is a convenience sample. Please describe the sample size calculation with the appropriate parameters.

(PRESENT REVIEW) Line 89: Despite not providing a sample size calculation, the authors fundamented their sample sizes using a recent and valid literature, which is acceptable.

Thank you.

(PREVIOUS REVIEW) Line 87: The selection of the TMD cases was performed using the DC/TMD, were the controls also selected using the same DC/TMD clinical history and examination performed in the TMD cases?

(PRESENT REVIEW) Line 100: In this version, it is clear that the controls also underwent the DC/TMD, which was not clear in the previous version.

Thanks.

(PREVIOUS REVIEW) Line 89: Patients 18 years old or older were selected, but this will include elderly population usually not affected by TMD. I believe that the age range between 18 and 50 years old (young adults) would have been selected, giving external validity to the results.

(PRESENT REVIEW) Line 95: I believe that a statement about the large age range should be included in the discussion section as a limitation of the study, considering that the sample has already been selected and cannot be changed (minor correction).

Thank you very much for this observation. The limitation has been included in lines 344-345.

(PREVIOUS REVIEW) Line 90: Which severe neurological and psychiatric pathologies were used as excluding criteria, please describe? How about medication affecting the central nervous system, and pain perception, such as anti-depressants?

(PRESENT REVIEW) Line 98: In this version, the neurological and psychiatric pathologies and the medication used which affected the CNS were described.

Thank you for your suggestion.

(PREVIOUS REVIEW) Line 92: same comments as in line 87 (see above)

Thank you.

(PREVIOUS REVIEW) Line 108: Was the examiner in charge of the clinical examination blinded to the results of the questionnaire on symptoms? Were these examiners calibrated in anyway, trained is not enough?

(PRESENT REVIEW) Line 116: In this version, it is clear that the clinical examiner was blinded to the questionnaire on symptoms of TMD. However, the examiners were not calibrated, and another statement in the discussion should be added (minor correction) as a limitation of the study.

Thank you for the suggestion. This point is included in limitation section, line 347-348.

Results

(PREVIOUS REVIEW) Line 197: Table 1: The mean age of both TMD cases and controls seems to be higher than reported in specialized literature (between 20 to 40 years old), please see comments on line 89. In addition, the possible socio-demographic differences seen in Table 1 between cases and controls (i.e., age, weight, particularly the gender proportions - 92% versus 59.6%, and others) must be tested statistically to see if there are differences. When statistically differences are found, they must be taken into account in the multivariate analysis. I believe that possible confounders were neither identified nor analyzed.

(PRESENT REVIEW) Considering that this is a reliability and validity study, I believe the control of confounders is not essencial, so it is acceptable at the present form.

Thank you.

(PREVIOUS REVIEW) No results were presented regarding the diagnostic TMD groups in which the selected TMD cases were diagnosed: muscle disorders, disc displacements, arthralgia, etc.

(PRESENT REVIEW) Line 201: It is clear now in this version that the diagnostic TMD groups were Ia and Ib of the DC/TMD.

Thank you.

Discussion

(PREVIOUS REVIEW) Line 275: The authors identified a higher proportion of women in the TMD cases, but make no reference about the huge differences between cases and the control group, or the measures to control this variable in the analysis stage.

(PRESENT REVIEW) Line 294: The authors in this version the authors left clear this limitation of the study regarding the gender differences between cases and controls, but again, considering that this a reliability and validity study, it is not also essencial.

Thank you.

(PREVIOUS REVIEW) Line 279: According to the authors: “No notable differences were found for to age, weight, height, BMI, average family income, or alcohol consumption”. Again, how can the authors possibly make that assumption, considering that no statistical test was used to compare TMD cases versus controls in Table I. Again, if detected, this identified confounder should have been handled in the statistical analysis.

(PRESENT REVIEW) Table 1: The authors added in this version the statistical tests and the p-value. Once more, considering that this is not a etiology study, this limitation does not affect the results.

Thank you.

(PREVIOUS REVIEW) Line 294: The authors stated: “In general, the results of the previous study appear better than those obtained in ours. This may be because Fernandes Pires' sample set contained patients who were on average 25 years younger than ours (47–48 years) and was composed only of women with muscle disorders (11), while ours also included men.” As already pointed out in previous comments, considering that the authors were trying to replicated the SFAI results in the Portuguese language (Fernandes previous study), it would have been more coherent to select the same age range, gender distribution, and TMD diagnostic groups of the previous reference study. Why did the authors make a different selection of cases and controls?

(PRESENT REVIEW) Line 315: In this version, the authors pointed out the differences between the two studies (Fernandes and this one), which was not clear in the previous version.

Thank you.

THIS IS THE REVIEW!

The authors really thank the reviewer for the comments, which have improved the quality of this work.